# Pre-aged terrigenous organic carbon biases ocean ventilation-age reconstructions in the North Atlantic

Jingyu Liu[1,2,9], Yipeng Wang[1,2,9], Samuel L. Jaccard[3,4], Nan Wang[5], Xun Gong [6,7], Nianqiao Fang[8] & Rui Bao [1,2] ✉

Changes in ocean ventilation have been pivotal in regulating carbon sequestration and release on centennial to millennial timescales. However, paleoceanographic reconstructions documenting changes in deep-ocean ventilation using $^{14}C$ dating, may bear multidimensional explanations, obfuscating the roles of ocean ventilation played on climate evolution. Here, we show that previously inferred poorly ventilated conditions in the North Atlantic were linked to enhanced pre-aged organic carbon (OC) input during Heinrich Stadial 1 (HS1). The $^{14}C$ age of sedimentary OC was approximately $13,345 \pm 692$ years older than the coeval foraminifera in the central North Atlantic during HS1, which is coupled to a ventilation age of $5,169 \pm 660$ years. Old OC was mainly of terrigenous origin and exported to the North Atlantic by ice-rafting. Remineralization of old terrigenous OC in the ocean may have contributed to, at least in part, the anomalously old ventilation ages reported for the high-latitude North Atlantic during HS1.

Rapid changes in ocean-circulation dynamics during the last deglaciation may provide fundamental constraints for understanding the past and predicting future variations in atmospheric $p$CO$_2$[1,2]. Many paleoceanographic reconstructions are consistent with a more poorly ventilated high-latitude deep North Atlantic Ocean during the last ice age and early deglaciation, with ventilation ages typically ranging between ~1000 and ~5000 yr (the benthic-atmosphere radiocarbon age offset)[3], significantly exceeding modern values (~400 yr)[4]. Reduced ventilation-defined here as the degree of air-sea equilibration - in the North Atlantic has commonly been attributed to dynamic changes in the Atlantic Meridional Overturning Circulation (AMOC) and the northward advection of poorly ventilated water masses from the Southern Ocean[5,6]. When pronounced local ventilation anomalies are taken into consideration, such as ventilation ages exceeding 5000 years in the intermediate-depth eastern equatorial Pacific, the dominant paradigm involving the advection of poorly ventilated water masses becomes, however, potentially inconsistent[7,8]. Indeed, $^{14}C$-depleted (local) carbon sources, such as inputs of $^{14}C$-dead hydrothermal carbon to the abyssal ocean[7,9], may account for some of the old ventilation ages inferred based on marine authigenic carbonate phases[7]. As changes in ocean circulation dynamics have been pivotal in modulating abrupt climate variability since the Last Glacial Maximum (LGM)[10–12], it is essential to understand the mechanisms that may have caused changes in ocean ventilation in the past and in particular across the last glacial termination.

[1]Frontiers Science Center for Deep Ocean Multispheres and Earth System, Key Laboratory of Marine Chemistry Theory and Technology, Ministry of Education, Ocean University of China, Qingdao 266100, PR China. [2]Laboratory for Marine Ecology and Environmental Science, Pilot Qingdao National Laboratory for Marine Science and Technology, Qingdao 266237, PR China. [3]Institute of Earth Sciences, University of Lausanne, Lausanne CH–1015, Switzerland. [4]Oeschger Center for Climate Change Research, University of Bern, Bern CH–3012, Switzerland. [5]Frontiers Science Center for Deep Ocean Multispheres and Earth System, Key Lab of Submarine Geosciences and Prospecting Techniques, Ministry of Education and College of Marine Geosciences, Ocean University of China, Qingdao 266100, PR China. [6]Institute for Advanced Marine Research, China University of Geosciences, Guangzhou, PR China. [7]Shandong Provincial Key Laboratory of Computer Networks, Qilu University of Technology (Shandong Academy of Sciences), Jinan, PR China. [8]School of Ocean Sciences, China University of Geosciences (Beijing), 100083 Beijing, PR China. [9]These authors contributed equally: Jingyu Liu, Yipeng Wang. ✉e-mail: baorui@ouc.edu.cn

Paleoceanographic evidence gleaned through the testimony of Northwestern Atlantic sediment cores indicates that catastrophic events episodically occurred during the last deglaciation, including outburst floods[13], abrupt fluvial runoff[14], and ice-rafting events[15,16]. The most prominent of these perturbations relate to Heinrich Events (HEs), during which large amounts of freshwater and terrigenous material[13–16] surged to the high-latitude North Atlantic and were discharged through icebergs[17], which transiently altered the intensity of the AMOC[18] and led to abrupt cooling in the northern hemisphere[15,16]. Modern oceanographic studies indicate that melting glaciers and ice sheets export large volumes of dissolved organic carbon (DOC, $1.04 \pm 0.18$ Tg·yr$^{-1}$, $1$ T $= 10^{12}$) and particulate organic carbon (POC, $1.97$ Tg·yr$^{-1}$) to the ocean[19], with both fractions being prone to bacterial degradation[20,21]. Given the widespread influence of HEs in the high-latitude North Atlantic[15,22], the impact of the pre-aged terrigenous OC supply and subsequent remineralization may significantly have affected the regional (radio)carbon inventory. Enhanced terrigenous OC supply associated with HEs could thus provide a potential source of $^{14}$C-depleted carbon to the ocean and account for, at least in part, the anomalously old ventilation ages reported for the high-latitude North Atlantic during the last deglaciation.

In this study, we measured the $^{14}$C ages of sedimentary OC and paired planktic and benthic foraminifera in three North Atlantic sediment cores (Sites U1302, 3560 m; U1308, 3870 m; and U1314, 2820 m; Fig. 1a and Methods) to investigate the impact the remineralization of pre-aged terrigenous OC imposed on North Atlantic ventilation age reconstructions across the last glacial termination. These three sites are located in the western, central, and northern North Atlantic, respectively, enabling us to investigate the spatiotemporal relationships between pre-aged terrigenous OC input associated with ice-rafting events and the evolution of subsurface ocean ventilation. Our data indicate that a large amount of terrigenous OC was exported to the North Atlantic during Heinrich Stadial 1 (HS1) and - to a lesser extent - the Younger Dryas (YD), which was substantially older than co-occurring foraminifera. The pre-aged terrigenous OC input was directly coupled with a decline in subsurface ocean ventilation inferred

based on the $^{14}$C age difference between coeval benthic-planktic (B-P) foraminifera in the central North Atlantic. Thus, our findings provide a perspective for understanding the multi-facetted mechanisms underlying reduced ocean ventilation in the high-latitude North Atlantic and their implications for the oceanic carbon cycle.

## Results and discussion
### Enhanced old terrigenous OC input to the North Atlantic during HS1
The $^{14}$C ages of sedimentary OC vary between $18,450 \pm 150$ and $30,200 \pm 310$ yr during HS1. These $^{14}$C ages are significantly older than those of co-occurring foraminifera, which typically range between $13,000 \pm 50$ and $18,950 \pm 500$ yr (Fig. 1b). On average, sedimentary OC is thus $8,564 \pm 305$ yr older than planktic foraminifera and $6,650 \pm 478$ yr older than benthic foraminifera during HS1 (Supplementary Dataset S2). Beyond HS1, the age difference between OC and biogenic carbonate phases is much more subdued, with sedimentary OC on average only $2,782 \pm 174$ yr older than coeval foraminifera (Fig. 2b, Supplementary Dataset S2). Furthermore, the $\delta^{13}C_{org}$ and $\delta^{15}N_{org}$ values at sites U1302 and U1308 are significantly lower during HS1, in line with the maximum C/N ratio (Supplementary Figs. 6 and 7), consistent with increased sedimentary burial of terrigenous OC in the western and central North Atlantic[23]. Microscopic observations suggest that the sediment layer is enriched with ice-rafted debris (IRD) (Supplementary Fig. 9), indicating that OC is intimately related to ice-rafting events characteristic of HS[15,16,24]. Rocks and sediments underlying the Laurentide ice sheet have been suggested to provide the main sources of IRDs in the western and central North Atlantic[15–17] (Fig. 1a). Site U1314 is located south of Iceland, implying that the northern North Atlantic may also have received terrestrial material from Iceland, Greenland, and/or Scandinavia[15]. Furthermore, our results indicate that the sedimentary OC is ~4000 yr older than the coeval foraminifera at Site 1314, suggesting that pre-aged terrigenous OC may also have been transported to the northern North Atlantic (Fig. 2b). While pilot studies reported that hydrodynamic processes could lead to OC aging about thousands years in the relatively-shallow oceans[25–27] and

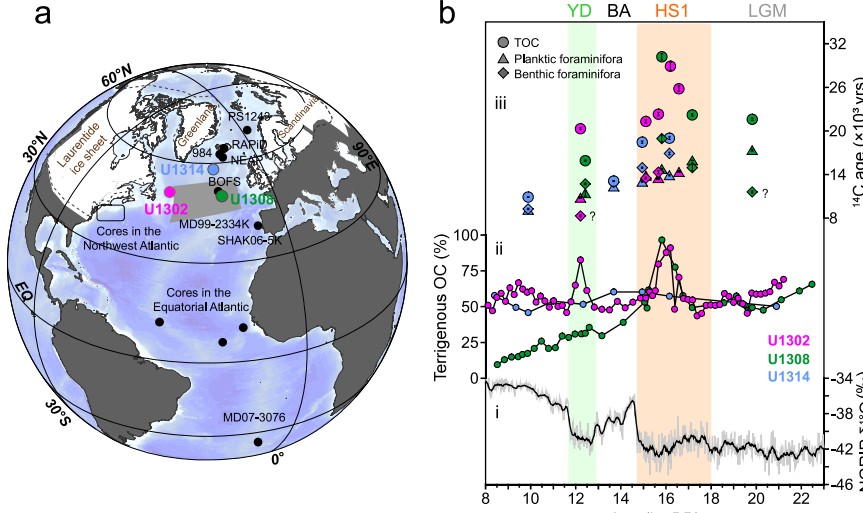

Fig. 1 | **Enhanced pre-aged terrigenous OC input to the North Atlantic during Heinrich Stadial 1 (HS1) and the Younger Dryas. a** Map showing the core sites considered in this study (Supplementary Table 1)[4,5,11,37–40,43,51,61]. The grey shaded area represents Ruddiman's IRD belt[15,16]. The dashed lines indicate the maximum ice sheet extent at the LGM[67]. **b i** North Greenland Ice Core Project (NGRIP) δ$^{18}$O serves as a reference chronology[68], the grey line represents the raw data and the black line reflects the 10-point running mean; ii sedimentary terrigenous Organic Carbon (OC) content (indicated by the δ$^{13}$C$_{org}$ data) at sites U1302 (magenta), U1308

(green), and U1314 (blue); iii $^{14}$C ages targeting the different sedimentary fractions, namely Total Organic Carbon (TOC, dots), planktic foraminifera (triangles), and benthic foraminifera (diamonds) at sites U1302 (magenta), U1308 (green), and U1314 (blue). The percentage of terrigenous OC is derived based on δ$^{13}$C$_{org}$, and the old OC is indicated by the older $^{14}$C age of TOC when compared to the co-deposited foraminifera. BA Bølling–Allerød, HS1 Heinrich Stadial 1, LGM Last Glacial Maximum, YD Younger Dryas. The earth map is drawn by Ocean Data View (Schlitzer, Reiner, Ocean Data View, odv.awi.de, 2023)[69].

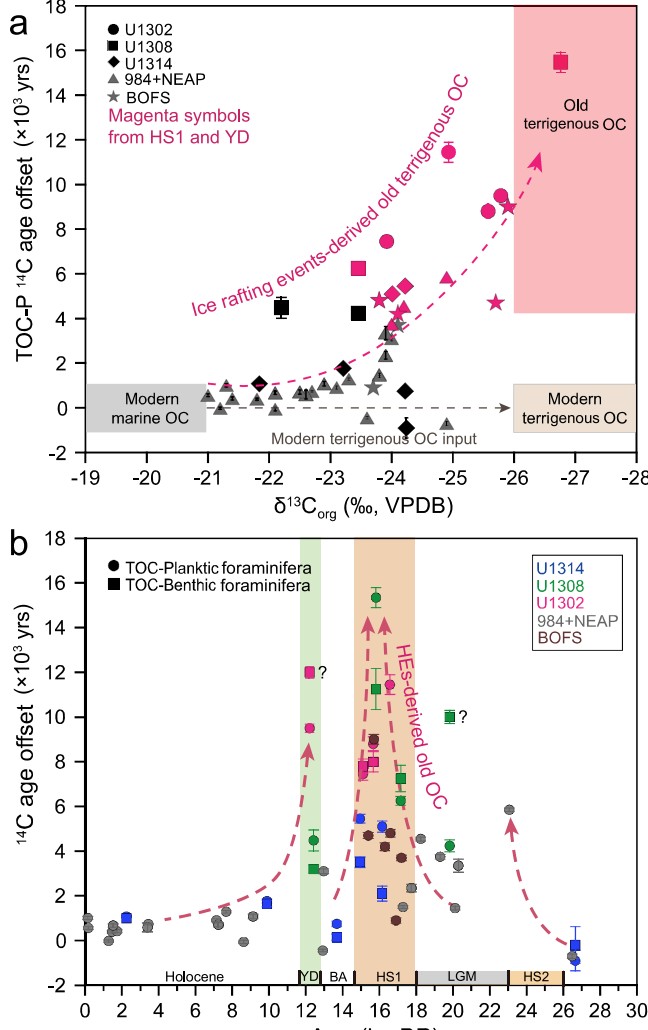

**Fig. 2 | Radiocarbon age analyses of TOC and foraminifera in the North Atlantic.** The data for sites U1302, U1308, and U1314 were generated in the context of this study; the data for sites 984, NEAP and BOFS were generated previously[38,61]. **a** Binary diagrams for $\delta^{13}C_{org}$ and $^{14}C$ offsets between Total Organic Carbon (TOC) and planktic foraminifera were used to identify the pre-aged, terrigenous OC fraction (with associated 1σ-uncertainties). The samples spanning HS1, HS2, and YD are indicated in red shading. **b** Simulated $^{14}C$ offset between TOC and benthic/planktic foraminifer versus sample age. Significant $^{14}C$ offsets between TOC and foraminifera were observed during HS1, HS2, and the YD (with associated 1σ-uncertainties), which indicate that pre-aged terrigenous OC input may have been derived from HEs. The few anomalous $^{14}C$ values of benthic foraminifera are indicated by question marks. HS1 Heinrich Stadial 1, HS2 Heinrich Stadial 2, YD Younger Dryas.

bioturbation may also affect the OC $^{14}C$ ages[28], it is difficult to interpret the such large $^{14}C$ age difference between OC and coeval planktonic foraminifera (~4240–15,440 yr) during HS1 in northern North Atlantic (More information see Supplementary Materials). To sum up, it is more reasonable that enhanced old terrigenous OC input is mainly attributed to the observed large $^{14}C$ offset need the contribution of pre-age OC remineralization.

## Remineralization of old terrigenous OC and its influence on ventilation ages

Our data reveal that the pre-aged terrigenous OC deposited during HS1, both in the central (U1308 Site) and northern (U1314 Site) North Atlantic, correspond to reduced ocean ventilation reconstructed

based on B-P $^{14}C$ ages with age differences ranging between 1950 ± 135 and 4190 ± 545 yr (Fig. 1b, Supplementary Figs. 6 and 8). As pre-aged terrigenous OC degrades, it may substantially alter the (radio)carbon inventory of ambient seawater[29], thereby influencing the $^{14}C$ ages of benthic foraminifera. Stable carbon-isotope evidence suggests that the deep North Atlantic was characterised by enhanced sequestration of remineralized carbon during the last deglaciation[29,30], consistent with the hypothesis that pre-aged terrigenous OC degradation may have occurred in the ocean interior.

The spatial heterogeneity in the B-P $^{14}C$ ages amongst the three sites highlights regional sedimentological characteristics (Fig. 1). In particular, the B-P $^{14}C$ age differences at Site U1308 indicate anomalously reduced ventilation in the central North Atlantic (4190 ± 545 yr, HS1). This may be related to the location of Site U1308, which lies within the IRD belt (Fig. 1), where the terrigenous OC input flux associated with IRDs may have been the highest[15,16]. While no significant increase in TOC burial was observed at Site U1308 during the deglaciation, enhanced supply of terrigenous OM has been reported previously based on biomarkers and reconstructions of past changes in sediment accumulation[31-33], suggesting that the remineralization of old OC may be intensive, impacting on ventilation age reconstructions at site U1308. Site U1314 is located in the northern reaches of the IRD belt, which was likely less affected by old terrigenous carbon input. This may have contributed to the comparatively lower ventilation ages at site U1314 when compared to site U1308. We did not observe reduced ventilation as inferred by $^{14}C$ age reconstructions during HS1 at Site U1302 (Fig. 1 and Supplementary Fig. 6), possibly owing to the high sedimentation rate (ranging between 16.5–17.3 cm/ka) and spatial differences of IRD sources[34]. The high sediment accumulation at U1302 may have prevented extensive remineralization of old OC due to its rapid burial and lower bacterial-respiration rates (i.e. reduced oxygen exposure time)[35,36] (Supplementary Fig. 6). Therefore, we suggest that the differential remineralization of old terrigenous OC may have contributed to the spatial heterogeneity in the ventilation ages reported for HS1 in the North Atlantic.

### Spatial Atlantic ventilation compilations

To further document the potentially obfuscating influence of old terrigenous OC remineralization on ventilation-age reconstructions, we compile the ventilation ages available for the entire North Atlantic basin (Fig. 3). Taking changes in surface reservoir ages into consideration, we hereafter report ventilation ages as the $^{14}C$ age difference between benthic foraminifera and the contemporaneous atmosphere (B-atm). Relatively well-ventilated intermediate-depth (<2000 yr) waters prevailed from low- to mid-latitudes[5,37] in the North Atlantic basin across the last deglaciation (Fig. 3c, e). In contrast, the high-latitude North Atlantic was characterised by generally more poorly ventilated conditions (with B-atm ~5000 yr) at both abyssal (Fig. 3f) and intermediate[38] (Fig. 3g) depths during HS1. This spatial pattern is difficult to reconcile with the prevailing notion that old (poorly ventilated) southern-sourced water (SSW) bathed large swaths of the deep North Atlantic at the end of the last ice age[5,39-41]. The ventilation ages of the SSW endmember typically range between ~2000 and 4000 yr[5,37] during HS1, somewhat younger than the old ventilation ages of the high-latitude North Atlantic (~5000 $^{14}C$ yr). Emerging evidence indicates that the ventilation of the South Atlantic, and more generally, the Southern Ocean, began to recover at the onset of HS1[5,37,42]. Would SSW have substantially intruded the deep North Atlantic during HS1, it would have resulted in ventilation ages (B-P $^{14}C$ ages) ranging between ~2000 and 4000 yr, which is arguably incompatible with a sluggishly ventilated high-latitude North Atlantic at that time.

The poorly ventilated Arctic-sourced water (ventilation ages up to 10,400 yr) may have contributed to reduce ventilation in the high-latitude North Atlantic[43] during the last ice age; however, whether it

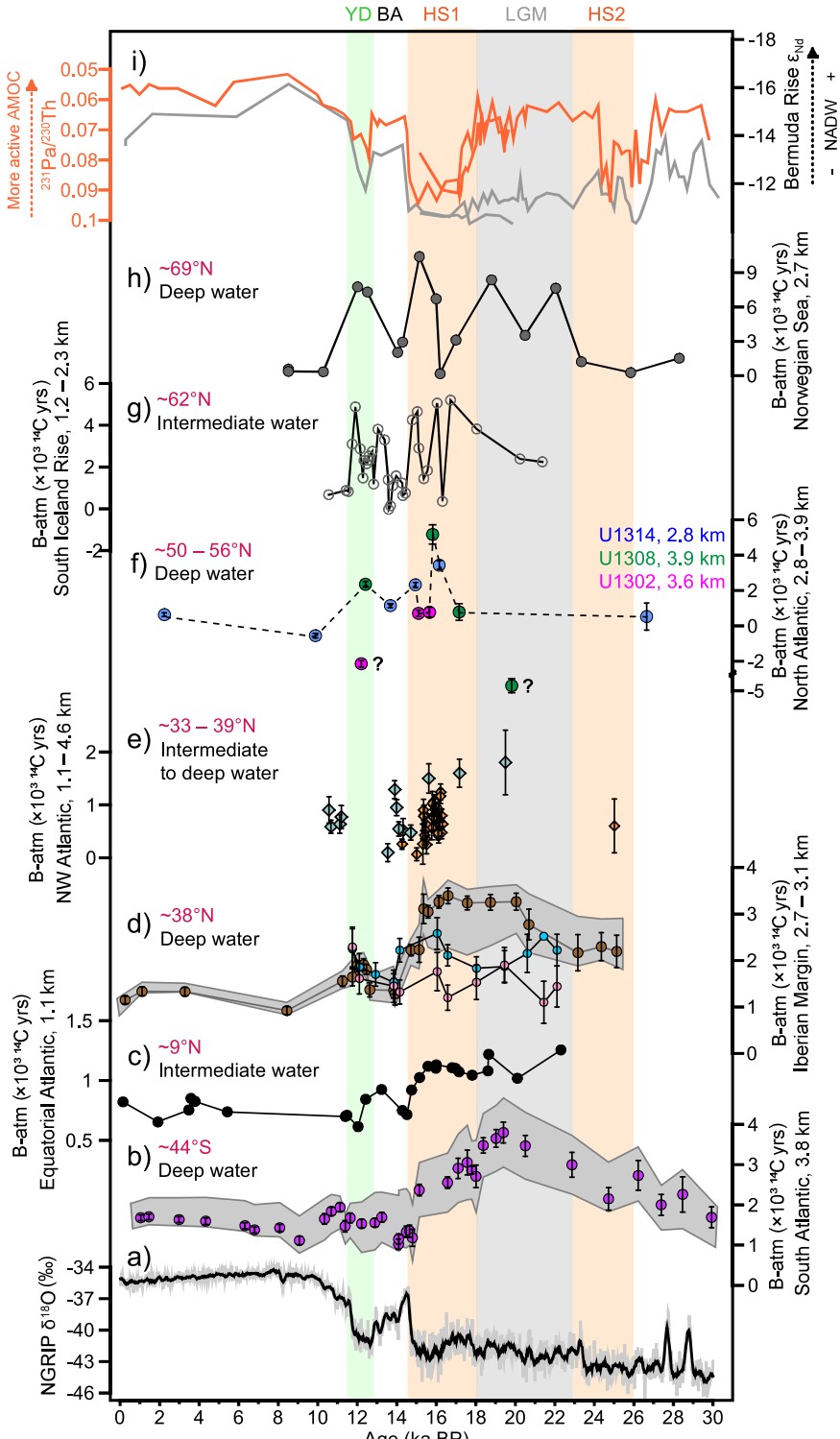

**Fig. 3 | Atlantic ventilation compilation coupled with circulation changes since the last ice age. a** Greenland Ice Core from the NGRIP δ18O[68]. **b** South Atlantic B-atm ages at water depth of 3.8 km[5,37], where the shaded area indicates the maximum/minimum range of B-atm offsets. **c** Coral-derived B-atm ages in the Equatorial Atlantic[11]. **d** B-atm ages at the Iberian Margin (data of Site SHAK06-5K (2.6 km, pink) and Site MD99-2334 (3.1 km, sky blue)[51]. The data for Site MD99-2334 (3.1 km, brown) were also reported previously[5,40]. The shaded area indicates the maximum/minimum range of B-atm offsets at site MD99-2334. **e** B-atm ages (coral-derived data, orange diamonds; benthic foram-derived data, light green diamonds) from the northwest Atlantic seamounts at water depths ranging 1.1–4.6 km[39]. **f** B-atm

ages of the high-latitude North Atlantic at 2.8–3.9 km (three sites in this study). The anomalous 14C values of benthic foraminifera are indicated by question marks. In term of the overall Atlantic ventilation compilation, we assembled our data from three sites to represent the situation of the high-latitude deep North Atlantic. **g** B-atm ages of the south Iceland Rise at depth of 1.2–2.3 km[3]. **h**, B-atm ages of the Arctic Mediterranean at a water depth of 2.7 km[43]. **i** 231Pa/230Th and εNd proxies showed variations of the AMOC and NADW, respectively. The orange lines indicate 231Pa/230Th data[17,44], and the grey lines indicate εNd data[45,46] measured at Bermuda Rise. All of the above have an associated 1σ-uncertainty.

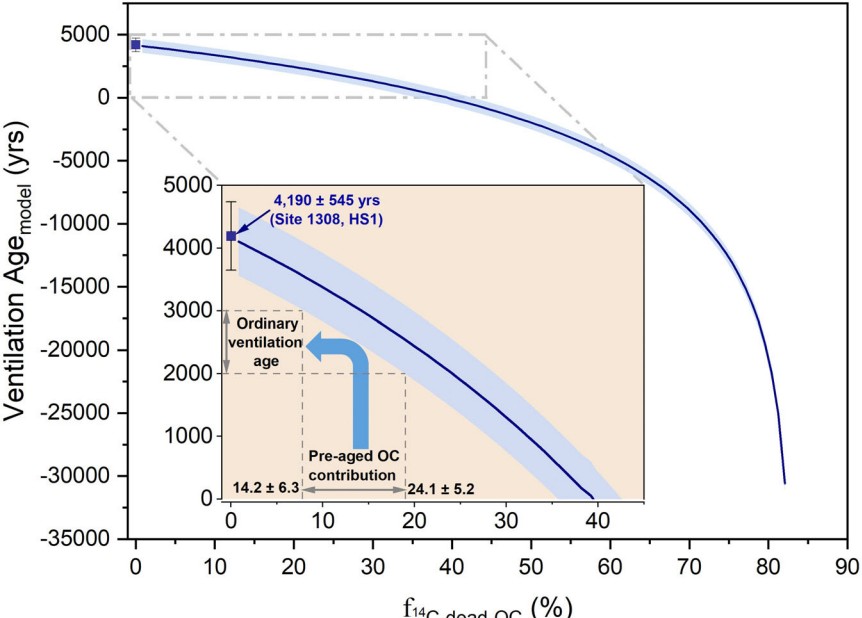

**Fig. 4 | Pre-aged OC contribution to benthic foraminifera calcite ($f_{14_{C-dead-OC}}$) during Heinrich Stadial 1 (HS1), versus modelled ventilation age (age_model).** [14]C ages of surface water and bottom water serve as a basis for developing the theoretical ventilation-age model described below (Methods; theoretical ventilation age). Pre-aged Organic Carbon (OC) contribution to benthic foraminifera ($f_{14_{C-dead-OC}}$) tended to negatively correlate with the ventilation age_model in the context of an old ventilation age (4190 ± 545 yr of B·P, with associated 1σ-uncertainties) at site U1308 during HS1. The results of the ventilation age_model were dramatically overprinted by pre-aged OC remineralization in the North Atlantic during HS1. The positive seawater-ventilation age trend can be theoretically tuned via remineralization of pre-aged OC (insert square). The ordinary ventilation age could be hundreds, or even a thousand years (extreme value up to 2000 yr)[3] may be achieved based on pre-aged OC inputs ranging between 14.2 ± 6.3% and 24.1 ± 5.2% $f_{14_{C-dead-OC}}$.

can also influence the deep North Atlantic remains questionable. To some extent, sedimentary [231]Pa/[230]Th and εNd evidence is consistent with a weaker AMOC state and reduced North Atlantic Deep Water (NADW) formation during HS1[17,44–46] (Fig. 3i), which may suggest that the contribution of overflow water from the Arctic Mediterranean to the North Atlantic remained limited. Furthermore, millennial-scale variations in ocean ventilation in the (sub)polar North Atlantic (Fig. 3f–g) and Arctic Mediterranean (Fig. 3h) coincided with those reported in the mid- (Fig. 3d,e) and low-latitudes North- (Fig. 3c) and South Atlantic (Fig. 3b), albeit to a much smaller magnitude. Such abrupt variations in ventilation ages challenge the mechanisms proposed to account for water mass advection from the south.

Assuming that the input of terrigenous OC to the high-latitude North Atlantic via ice-rafting amounted ~50 Tg·yr⁻¹ during HS1 – 50 times the OC-input flux by the modern Greenland Ice Sheet (~1 Tg·yr⁻¹)[18,20] – sustained for ~1500 yr[47], we estimate that ~75 Pg (1 Pg = 10[15]g) of terrigenous OC may have been exported to the North Atlantic during HS1. We estimate that at least ~18 Pg terrigenous OC would have been remineralized in the North Atlantic during HS1. Thus, the remineralized OC may have significantly impacted the [14]C ages of benthic foraminifera ([14]C_Benthic), which are commonly used for ventilation-age reconstructions. Based on our semi-quantitative framework (Method and Fig. 4), assuming that ~10.7–18.1 Pg pre-aged terrigenous OC was remineralized and efficiently assimilated by benthic foraminifera during HS1, the overestimation of ventilation age may amount up to thousand years during HS1. Furthermore, marine-sourced, labile OC may be pre-aged in the water column before deposition[48]. The aged marine OC degradation may further accelerate the remineralization of pre-aged terrigenous refractory OC[49,50]. Considering the duration of the events, the maturation and degradation of marine OC and its potential impact on ventilation age reconstructions may be a target for research in the future. Hence, researchers should not ignore the influence of pre-aged remineralized OC on reconstructed ventilation ages[5,37,40,51].

In addition, the terrigenous OC supplied by ice-rafting to the high-latitude North Atlantic may have provided a hitherto unaccounted for source of carbon to the atmosphere during the last deglaciation. Were the respired carbon derived from the remineralization terrigenous OC released to the atmosphere, then it may have contributed, at least in part, to the rapid rise in atmospheric $CO_2$ concentrations[1]. HEs or Heinrich-like events have been proposed to have occurred during each glacial termination, since at least ~640 ka BP[16]. It is therefore possible that the terrigenous OC input associated with HEs or Heinrich (-like) events and its subsequent remineralization and release may have contributed to the rapid rises in atmospheric $CO_2$ that characterised previous glacial terminations[52]. Further investigations are needed to better quantify the impact of catastrophic outflow events on ocean ventilation and circulation during deglaciations and their implication for the global carbon cycle.

## Methods
### Sites and samples
Integrated Ocean Drilling Program (IODP) Site U1314 is located on the southwest flank of the Reykjanes Ridge (Iceland) on the southern Gardar Drift (2820 m, 56°21.89′N, 27°53.28′W) (Fig. 1). The Gardar Drift is largely influenced by the lower part of the AMOC, the Iceland-Scotland Overflow Water (ISOW), or the Northeast Atlantic Deep Water (NEADW) and seafloor topography[53]. Site U1308 (3870 m, 49°52.67′N; 24°14.28′W) is located in the central North Atlantic and Ruddiman's IRD belt, making it a key site for investigating ocean–atmosphere–ice sheet interactions, which have a profound impact on global climate change. Site U1302 (3560 m, 50°10.01′N; 45°38.32′W) is located on the crest of a small ridge, east of a fault scarp marking the eastern side of Orphan Knoll. Site U1302 is bathed by Labrador Sea Water which may transport terrestrial materials from the North American continent[54,55].

At these sites, IRDs were commonly deposited during the last deglaciation. We sampled the upper 3.12 mcd composite core of U1314B/C at 10 cm resolution, the upper 1.40 mcd of U1308B at 2 cm

resolution, and the upper 3.76 mcd of U1302D/E at 4 cm resolution (which spans the last ~30 ka BP, thousand years before present), in order to investigate to which degree old terrigenous OC input may have affected $^{14}$C-based ocean ventilation reconstructions in the North Atlantic.

## Age model

The age models developed in this study were primarily based on $^{14}$C dating of planktonic foraminifera in the three IODP cores. The age model used for U1314 was based on seven planktic foraminifera (*G. bulloides*) $^{14}$C dates and Bacon modelling[56]. The reservoir ages and standard errors at Site 1314 were obtained from the literature[57]. The stratigraphy for site U1308 used here was also published previously[47], with 24 planktic foraminifera (*G. bulloides*) used for $^{14}$C dating at sites U1308A[47] and 609[58]. We recalculated calendar ages based on the latest Marine20 curve and Bacon modelling[59]. We used magnetic suscept-ibility to graphically align U1308A, U1308B, and U1308C and propose a revised metre composite depth (rmcd) model. The age model for U1302 was based on four planktonic foraminifera (*N. pachyderma*) $^{14}$C dates. Owing to the lack of constraints on the reservoir age during the last deglaciation in the vicinity of site U1302, modern ΔR values were used to determine calendar ages using the Bacon model. See the Supplementary Information for additional details (Supplementary Figs. 1 and 4).

## TOC, TN, and $δ^{13}C_{org}$, $δ^{15}N_{org}$ analysis

The samples were freeze-dried, weighed (0.5 g), homogenised, and acidified using 1 M HCl to remove carbonate until no bubbles were present. The samples were then rinsed repeatedly with Milli-Q water until the pH reached neutrality, after which the samples were freeze-dried again. All glassware used was pre-combusted at 450 °C for 5 h prior to use. Carbon and nitrogen content and isotope analyses were performed with an EA-IsoLink elemental analyser coupled to a MAT253 plus isotope ratio mass spectrometer via a ConFlo IV universal inter-face (all from Thermo Fisher Scientific, Bremen, Germany) at the Key Lab of Submarine Geosciences and Prospecting Techniques (Ministry of Education, Ocean University of China). The $δ^{13}C_{org}$ data are reported relative to the Vienna Pee Dee Belemnite (V-PDB) standard with an external analytical precision of ±0.1 ‰, and the $δ^{15}N$ data are reported relative to air with an external analytical precision of ±0.5 ‰. The external analytical precisions of the TOC and TN data were ±5 ‰ and ±10 ‰, respectively.

## $^{14}$C analysis

In this study, we measured the $^{14}$C contents of TOC and foraminifera at the National Ocean Sciences Accelerator Mass Spectrometry Facility at Woods Hole Oceanographic Institution (WHOI). Acid pre-treatment was conducted at the WHOI. Planktonic foraminifera (*Globigerina bulloides* and *Neogloboquadrina pachyderma* (sin.)) were subjected to five interations of ultrasonic treatment in Milli-Q water to remove excess clay minerals[60], as shown in Supplementary Dataset S1. The benthic foraminifera for conducting $^{14}$C analysis included *Cibicidoides wellerstorfi, Uvigerina* spp. *and Melonis* spp.

## Source apportionment of OC

The following two-end-member model (terrigenous and marine OC) for $δ^{13}$C was used to identify the OC source (Supplementary Data-set S3−S5):

$$δ^{13}C_{TOC} = f_{terr} × δ^{13}C_{terr} + f_{mar} × δ^{13}C_{mar} \quad (1)$$

$$f_{terr} + f_{mar} = 1 \quad (2)$$

where $δ^{13}C_{TOC}$ is the $δ^{13}$C of bulk sediments and $f_{terr}$ and $f_{mar}$ represent the relative proportions of terrigenous OC and marine OC, respec-tively. The $δ^{13}C_{terr}$ value was assigned to an average value of −27‰, reflecting the −26 to −29‰ $δ^{13}$C values of TOC and the −26 to −30‰ $δ^{13}$C values for Phanerozoic OC-rich rocks underlying these soils[61,62]. The $δ^{13}C_{mar}$ endmember value amounts to −20‰, representative of the values reported previously[61,63].

Furthermore, the bulk sedimentary $δ^{15}$N and C/N values were used to constrain terrigenous OC input. According to previous studies, C/N ratios >10[64] and $δ^{15}$N values of <1‰[65] are diagnostic for a terrigenous provenance for the OC.

## Ventilation by B-P foraminifera and the B-atm

In this study, two methods were used to quantify past changes in ocean ventilation[40], namely B-P foraminifera age differences and the $^{14}$C off-set between benthic foraminifera and the contemporary atmosphere (B-atm). To investigate the potential contrasts in ventilation age reconstructions among the three sites in this study, we used the B-P approach to avoid the uncertainties related to constraining calendar ages. When comparing ventilation ages across the entire Atlantic, the B-atm approach was used to unify the available datasets and overcome the drawback pertaining to the lack of planktonic data, which was reported as a limitation in previous studies[3,11,39,43]. In this study, when considering the western North Atlantic[39], we used atmospheric $^{14}$C ages derived from IntCal20[59] (see the data in Supplementary Dataset S1 and S2 for further details). We refer to the original data from the lit-erature to obtain the ventilation ages mentioned in the main text. We believe that the influence of using different IntCal datasets on venti-lation age reconstructions remained within the normal error range (generally <500 $^{14}$C yr) and did not affect our argumentation.

## Theoretical ventilation age

First, two end-member models (OC and bottom water) for $Δ^{14}$C were applied to quantify the $^{14}$C contribution from OC to the benthic for-aminifera

$$Δ^{14}C_{Benthic} = f_{OC} × Δ^{14}C_{OC} + f_{BW} × Δ^{14}C_{BW} \quad (3)$$

$$f_{OC} + f_{BW} = 1 \quad (4)$$

where $Δ^{14}C_{Benthic}$, $Δ^{14}C_{OC}$, and $Δ^{14}C_{BW}$ represent $Δ^{14}$C values derived from benthic foraminifera, OC, and bottom water, respectively; $f_{OC}$ and $f_{BW}$ represent the relative $Δ^{14}$C proportions of OC and the bottom water to benthic foraminifera. Based on the $^{14}$C age offsets between the OC and benthic/planktic foraminifers shown in Fig. 2B, we selected values for $Δ^{14}C_{Benthic}$ (−905.8%) and $Δ^{14}C_{OC}$ (−976.9%) to analyse the relationship between $f_{OC}$ and $Δ^{14}C_{BW}$. These values for $Δ^{14}C_{Benthic}$ and $Δ^{14}C_{OC}$ were defined at Site U1308 where the ventilation ages were the oldest (4,190 ± 545 yr, B-P $^{14}$C age).

Second, we substituted Eqs. (3) and (4) to obtain the relationship between $f_{OC}$ and $Δ^{14}C_{BW}$, using Eq. (5).

$$Δ^{14}C_{BW} = (Δ^{14}C_{Benthic} − f_{OC} × Δ^{14}C_{OC})/(1 − f_{OC}) \quad (5)$$

The theoretical ventilation age (B-P $^{14}$C age) was considered as representing the offset in $^{14}$C age between bottom and surface waters, and Age$_{SW}$ was considered a fixed value for the $^{14}$C age of planktonic foraminifera during HS1.

$$\text{Ventilation Age}_{model} = \text{Age}_{BW} − \text{Age}_{SW} \quad (6)$$

Where, $Age_{BW}$ and $Age_{SW}$ represented the $^{14}C$ age of the bottom water and surface water, repsectively. $Age_{BW}$ would be expressed as follows:

$$Age_{BW} = -8033 \times \ln(Fm_{BW}) \tag{7}$$

Thus, $\Delta^{14}C_{BW} = \{Fm_{BW} \times e^{[0.00012097*(1950-2006)]} - 1\} \times 1000 \tag{8}$

$Fm_{BW}$ was the Fm value of bottom water.

By substituting Eqs. (5), (7), and (8) into Eq. (6), we obtained Eq. (9) that related the ventilation $age_{model}$ and $f_{OC}$ values

$$\text{Ventilation Age}_{model} = -8033 \times \ln\{[(\Delta^{14}C_{Benthic} - f_{OC} \times \Delta^{14}C_{OC}) \\ /(1-f_{OC})/1000+1]/e^{[0.00012097 \times (1950-2006)]}\} - Age_{planktonicHS1} \tag{9}$$

Here, $Age_{planktonicHS1}$ was a fixed value for the $^{14}C$ age of planktonic foraminifera during HS1.

Finally, we resolved the relationship between $f_{OC}$ and $f_{14_{C-dead-OC}}$ during HS1 to determine the function relating ventilation the $Age_{model}$ and $f_{14_{C-dead-OC}}$ values ($^{14}C$-dead contribution of OC to benthic foraminifera; Fig. 4).

To quantify the amount of pre-aged OC exported into the North Atlantic that caused the largest ventilation age during HS1 at U1308, we adopted Eqs. (10) and (11):

$$\Delta^{14}C_{OC} = f_{deadcarbon} \times \Delta^{14}C_{deadcarbon} + (1 -_{deadcarbon}) \times \Delta^{14}C_{AgeHS1} \tag{10}$$

$$f_{14_C-dead-OC} = f_{OC} \times f_{deadcarbon} \tag{11}$$

$\Delta^{14}C_{OC}$, $\Delta^{14}C_{dead\,carbon}$ and $\Delta^{14}C_{AgeHS1}$ represented fixed $\Delta^{14}C$ dates for OC, $^{14}C$ dead carbon, and the $^{14}C$ age model, respectively; $f_{dead\,carbon}$ represented the percentage of dead $^{14}C$ in OC.

We derived that OC was composed of 82.9% dead carbon during HS1.

Furthermore, we estimated the pre-aged OC flux based on Eq. (12):

$$\text{Pre} - \text{aged OC flux} = \text{OC flux} \times f_{14_C-dead-OC} \tag{12}$$

where the OC flux was estimated as described above in the main text as 75 Pg in the North Atlantic during HS1; we estimated that the pre-aged OC flux ranged from 10.7 to 18.1 Pg.

## Data availability

The data generated in this study are provided in the Supplementary Information and Figshare data repository (https://doi.org/10.6084/m9.figshare.22634719)[66].

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

## Acknowledgements
This study was supported by the National Natural Science Foundation of China (grant numbers 92058207, 42076037, and 42242601, B.R.) and Fundamental Research Funds for the Central Universities (grant number 202042010, B.R.). This study was also funded by the Taishan Young Scholars (grant number tsqn202103030, B.R.) and the Shandong Natural Science Foundation (grant number ZR2021JQ12, B.R.). We appreciate the IODP community for sampling assistance on cruises and in laboratories.

## Author contributions
R.B. and J.L. designed the research; J.L. and N.W. performed the laboratory analysis; R.B., J.L., and Y.W. analysed the data and wrote the paper. R.B., J.L., Y.W., S.J., X.G., and N.F. contributed to the interpretation of results and commented on the final manuscript.

## Competing interests
The authors declare no competing interests.
