## [Peer Review File NEW · Nature Communications]

Pre-aged terrigenous organic carbon biases ocean ventilation-age reconstructions in the North AtlanticReviewer #1 (Remarks to the Author):

This is an interesting paper. It interprets the much older B-A age of subpolar North Atlantic deep water than subtropical and tropical Atlantic as caused by the pre-aged terrestrial OC input by ice rafting after remineralization. As a modeler, I am not familiar with the literature and is unable to judge if this point has been raised before in the community of marine carbon dating. If not, this work is novel and should be published, after some minor revision.

Comments:

1: Is this mechanism particularly effective for subpolar North Atlantic (because it relies on ice rafting)? If yes (my understanding), the paper title should be explicit on "North Atlantic".

2: L228: "Based on our semi-quantitative framework, (Method and Fig. 4), assuming that ~10.7 229 – 18.1 Pg pre-aged terrigenous OC was remineralized and efficiently assimilated by benthic 230 foraminifera during HS1, one could reasonably account for the ventilation ages reported for the North Atlantic during HS1." How much older? Please quantify in years.

Is this quantification assuming the remineralized carbon all in the subpolar North Atlantic? Is it possible part of the remineralized terrigenous OC been exported to lower latitudes?

Reviewer #2 (Remarks to the Author):

Review of "Pre-aged terrigenous organic carbon biases ocean ventilation-age Reconstructions" by Liu et al.

This Manuscript investigates the role of terrestrial OC transported by IRDs on the estimation of deep ocean ventilation during Heinrich Stadial1 in the North Atlantic Ocean. The authors use a wide array of OM-derived proxy records, like $\delta^{13}\text{C}$, $\delta^{15}\text{N}$, TOC%, C/N, and the ^{14}C age of (and corresponding ^{14}C age differences among) OC, planktic and benthic foraminifera. Analyses are performed in three cores located in the North and central Atlantic Ocean and retrieved within the IODP program.

The manuscript is well written, clear, and concise, and it reads very well. The theory to explain low ventilation ages at times of HS is novel and it delves into an aspect up to date overlooked by paleoceanographers. Despite the theory seems sound and plausible, the data, and their interpretation, are not entirely sufficient to provide robust evidence to support their hypothesis. In fact, I believe part of the presented data might be better explained by other mechanisms. Thus, there are some important aspects that should be addressed prior publication. Please find my major comments below:

The authors ascribe the age offsets between OC and planktic foraminifera solely and uniquely to the addition of terrigenous OC. I read the whole Manuscript waiting for them to mention and discard other possible processes that might lead to similar age offsets, but they simply neglect them. Based on which data, estimates, and other bibliography do the authors disregard the potential addition of pre-aged and fossil OC resuspended from continental slopes (i.e., marine OC) and bioturbation?

How do you explain the different magnitude of the OC-Planktic Foram age in cores that are at the same latitude and under the path of IRD belt? Specially, for BOF and 1308, which are located nearby? Measurement of duplicates, especially for core 1308, would be ideal.

Site U1308 is the one presenting the largest age difference between OC and planktic foraminifera (ca. 14 kyr). Samples from Site U1308 are from hole B. As shown in Hodell et al., 2017, digital image analysis in U1308B evidenced large traces fossils and a high bioturbation index for HS1.

Indeed, bioturbation has proven to have a large impact on the IRD record of U1308B by mixing the two distinct IRD peaks (H1.1 and H1.2.) observed in U1308A into one single peak during HS1. The authors overlook the potential impact of bioturbation on the data from this core. Bioturbation may partly explain (or even entirely?) the large observed age discrepancies.

In this regard, can you discard the impact of bioturbation on the other two cores? Additionally, sedimentation rates should be plotted on age-depth model figures to identify periods of low sedimentation rate when the impact of bioturbation may increase.

Line 124: "Microscopic evidence suggests that terrigenous OC is typically associated with ice-rafted debris (IRDs) (Extended Data Fig. 9)". How can the authors tell from these pictures? OC (and its origin, i.e., marine or terrestrial) is not discernible from optical microscope pictures. In this line, IRDs have low Mineral Surface Area (MSA) because of their large size. This means that IRD grains host very little OC in comparison with smaller mineral grains having higher MSA. Such small grains can be laterally transported by deep currents, introducing way larger amounts of pre-aged OC, typically marine. The magnitude of the age discrepancy between OC and planktic forams is quite large at some sites, meaning that a large amount of pre-aged/fossil OC needs to be added to induce such bias. The question is, given that IRDs have very low MSA, can IRDs host and introduce the large amounts of OC needed to induce the observed age discrepancies? Related> Line 147: So anomalous that a huge amount of fossil OC would be needed to explain it. Again, can IRDs transport so much OC given their low MSA? Please check my comment above.

Line 131: the authors state that the ca. 4000 yrs age difference between OC and planktic forams at site U1314 suggests pre-aged terrigenous OC. As shown in Extended Data Fig.8, C isotope and C/N records do not fully support the primarily terrestrial origin of OC input during HS1 at that site. Moreover, these proxy records remain constant prior, during, and after that period. Why would any potential terrestrial OC input influence ventilation ages only during HS1 and not before or after? Accordingly, the observed age discrepancy in core U1314 could be ascribed to the lateral transport of marine OC. Why the authors do not even mention the potential impact of other factors like bioturbation or, more likely, the addition of pre-aged marine OC at this site? The corresponding author has implemented excellent prior work dealing on the contribution of hydrodynamic transport of marine OC to foraminifera-OC age discrepancies... Therefore, the data indicate that there is not enough evidence as to state that "Our data reveal that the old sedimentary OC deposited during HS1, both in the central (U1308 Site) and northern (U1314 Site) North Atlantic...".

Line 150: can such a small peak in sedimentary TOC concentration be considered as significant? Given the vigorous bioturbation affecting core U1308B during HS1, and that the TOC flux at that time is as low as before and after the event, can such low ventilation ages be associated with an input of terrigenous OC associated with IRDs?

Line 155: "We did not observe reduced ventilation during HS1 at Site U1302". However, C isotope and C/N from U1302 do indicate high terrigenous OC input... Does this imply that the large input of terrestrial OC (as inferred from proxy records) at this site did not have any effect on ventilation ages but a smaller terrestrial input did it at other locations? The authors argue that this could be due to an increase in sedimentation rate, but they do not provide values.

Lines 183-185: The authors claim "In contrast, the high-latitude North Atlantic was characterised by generally more poorly ventilated conditions (with ~5,000 yrs of B-atm) at both abyssal (Fig. 3f) and intermediate (Fig. 3g) depths during HS1" But... AMOC stopped at that time, and ventilation of intermediate and deep water would decrease compared to the previous periods even without the input of terrigenous OC...

Extended Data Fig. 5. A 90% contribution of terrigenous OC seems a lot for sites so distant from the continents. Did the authors account for the degradation of more labile (i.e., marine) OC with time? It might be that the initial % of terrestrial OC was much lower, but increased with time due to the preferential remineralization of marine OC. What is measured is the remaining OC. In fact, if

the remaining buried OC is terrestrial, does not this mean that what was preferentially remineralized (and then influenced the radiocarbon signature of deep-water masses) was the more labile, marine OC?

Why is the YD not discussed at all?

Fig. 2 b: Data on cores U1308 and U1302 (the ones showing the largest OC-Planktic forams age discrepancies) are only available for HS1 and YD. Some data on other periods (e.g., Holocene, BA) would be desirable to see whether such large age discrepancies are characteristic of these periods or a relatively permanent feature of these sediments for the last deglaciation and Holocene.

In summary, my major concerns are related to the fact that the authors do not provide robust evidence for some of the statements they make or some of the conclusions they reach. The presented data could be partly or entirely explained by the influence of other processes (e.g., bioturbation (consistent with the evidence existing from core U1308 (Hodell et al., 2017)) and lateral transport of marine OC (consistent with the data provided by the authors for core U1314)). However, the authors do not provide any estimates of the potential influence of such processes. More worryingly, they do not even mention these nor discuss them on the text. If the influence of this processes cannot be discarded, one cannot unequivocally and exclusively ascribe low ventilation ages to the input of terrestrial OC.

More in detail:

-The authors claim reduced ventilation at Site U1314 and ascribe it to the input of terrigenous OC associated to IRDs, but terrigenous OC input seems constant in relation to previous and later periods according to various isotope proxy records.

-The authors claim reduced ventilation at Site U1308, which, to the best of my knowledge, is heavily affected by bioturbation (Hodell et al., 2017).

-The authors observe high ventilation at Site U1302, even negative values, which indicates something is affecting those ages (for which the authors give no satisfactory explanation). However, proxy records of terrigenous OC input at this Site do indicate an important input. They argue this could be due to high Sed. Rates, but values are not provided.

Minor comments:

Line 50: Does this refer to the surface or the deep reservoir?

Lines 54-56: This notion could be better explained, please rephrase.

Line 84: By "abnormal" do you mean "negative"? Please rephrase.

Fig. 1 ii) I think you mean OC%, since $\delta^{13}C$ is not shown. Please, change the figure or the text "The terrigenous OC is indicated by the $\delta^{13}C_{org}$ data".

Fig.3. "(data of Site SHAK06-5K (2.6 km, sky blue) and Site MD99-2334 (3.1 km, pink)". Records are reversed, the blue one corresponds to MD99-2334 and the pink one corresponds to SHAK06-5K.

Line 193: a word is missing. A poorly ventilated... (ocean, water mass?) in the North Atlantic.

Lines 214-215: That is not true. The graph shows millennial-scale variations in Fig. 3 d-e. You may want to say that those variations, which undoubtedly exist, are of a smaller magnitude than those found in the north.

Line 231-233: Could the authors develop on this idea? I don't follow their reasoning here.

Line 404: which benthic species?

414-415: Please update references by adding Verwega et al., 2021.

Blanca Ausín

Reviewer #3 (Remarks to the Author):

Liu et al. presents compelling new evidence from organic carbon ^{14}C signatures from sediment cores located in the high latitude North Atlantic to explain the poorly ventilated deep North Atlantic indicated by ^{14}C from benthic foraminifera during the last deglaciation. Overall, the results are convincing and the manuscript is generally well-structured.

My major comment is about the term "ventilation", which should be clarified in the manuscript. Ocean ventilation is controlled by the physical circulation/overturning rates and surface air-sea gas exchange. ^{14}C has been used to indicate ocean ventilation, but it is not a direct measurement of it. For example, the "reduced ventilation" in line 155 is actually "older ^{14}C age". The definition of ventilation, ventilation age, and ^{14}C age should be described in the manuscript to avoid confusion. Furthermore, this study attributes the old ^{14}C in the deep North Atlantic to the pre-aged terrigenous organic carbon. The authors claim that their results provide a unique perspective regarding ocean ventilation. However, this "perspective" is not well discussed in the manuscript. What is the ocean ventilation (physical circulation and air-sea gas exchange) during the deglaciation that can be inferred from the new OC ^{14}C results? The current manuscript only explains the ^{14}C age instead of ocean ventilation.

Another comment is the usage of B-P ^{14}C age to estimate ocean ventilation. Sites in this study are bathed by water sourced from the Southern Ocean. Using planktonic ^{14}C ages from the North Atlantic will bias the estimation of the ventilation age. Will this affect the conclusion claimed in this study?

Line 191-193: The authors rule out the SSW advection to the North Atlantic by comparing the ventilation ages between SSW and North Atlantic as the SSW is younger than the North Atlantic. It takes time for SSW to be advected to the North Atlantic, therefore the water age in the North Atlantic is still older than the Southern Ocean. So the older ventilation age in the North Atlantic than the Southern Ocean is still compatible with SSW being advected to the North Atlantic.

Response to Reviewers

We appreciate the constructive comments from the reviewers and editor concerning our manuscript entitled “*Pre-aged terrigenous organic carbon biases ocean ventilation-age reconstructions in the North Atlantic*” (Manuscript ID: NCOMMS-22-45880-T). These comments are very helpful for improving our manuscript. In this point-by-point *Response to Reviewers*, we have addressed all the comments and suggestions. The responses to comments from reviewers and editor are listed below, with comments listed first in *blue italics*, followed by our response in black. We hope that reviewers and editor are satisfied with our responses and the revised manuscript we provide.

Comments from Reviewer #1

This is an interesting paper. It interprets the much older B-A age of subpolar North Atlantic deep water than subtropical and tropical Atlantic as caused by the pre-aged terrestrial OC input by ice rafting after remineralization. As a modeler, I am not familiar with the literature and is unable to judge if this point has been raised before in the community of marine carbon dating. If not, this work is novel and should be published, after some minor revision.

Response: We appreciate your efforts in reviewing our manuscript and positive comments! Thank you very much for highlighting the novelty of our study. We believe that our investigations will shed new light on the complex, multi-faceted North Atlantic ventilation age reconstructions. Thank you very much for supporting publication of a revised version of our manuscript.

1: Is this mechanism particularly effective for subpolar North Atlantic (because it relies on ice rafting)? If yes (my understanding), the paper title should be explicit on “North Atlantic”.

Response: Thank you for the suggestion. In the revised manuscript, we updated the title, which now reads “*Pre-aged terrigenous organic carbon biases ocean ventilation-age reconstructions in the North Atlantic*”

2: L228: “Based on our semi-quantitative framework, (Method and Fig. 4), assuming that ~10.7 – 18.1 Pg pre-aged terrigenous OC was remineralization and efficiently assimilated by benthic foraminifera during HSI, one could reasonably account for the ventilation ages reported for the North Atlantic during HSI.” How much older? Please quantify in years.

Response: Thank you very much for your comments. Pioneering studies show that radiocarbon-based deep North Atlantic Ocean ventilation ages typically range between 2,000 yrs and 3,000 yrs during HS1 (e.g., Thornalley et al., 2011). Yet, our study implies a possible, yet unaccounted for, contribution of pre-aged terrigenous OC, transported to the open Atlantic Ocean by drifting icebergs, that would have substantially affected the radiocarbon-based ventilation age reconstructions. According to our calculations and assuming that 10.7 Pg to 18.1 Pg pre-aged OC were remineralized in the North Atlantic water column, the radiocarbon-based North Atlantic Ocean ventilation age reconstructions would have increased from several hundreds to thousands years. We updated sentence is at Lines 180-183 as follows:

“Based on our semi-quantitative framework (Method and Fig. 4), assuming that ~10.7 – 18.1 Pg pre-aged terrigenous OC was remineralized and efficiently assimilated by benthic foraminifera during HS1, the overestimation of ventilation age may amount up to thousand years during HS1.”

Reference:

Thornalley, D. J. R., Barker, S., Broecker, W. S., Elderfield, H. & McCave, I. N. The Deglacial Evolution of North Atlantic Deep Convection. *Science* **331**, 202–205 (2011).

3: Is this quantification assuming the remineralized carbon all in the subpolar North Atlantic? Is it possible part of the remineralized terrigenous OC been exported to lower latitudes?

Response: Thank you very much for pointing this out. Yes, we assume that the terrigenous organic matter, which was assumed to have been transported to open subarctic North Atlantic by ice rafting during HS1 (Naafs et al., 2013), was entirely remineralized locally. In this study, we highlight substantial differences in ventilation ages between the low- to mid-latitudes and high-latitude deep North Atlantic Ocean (Fig. 3), and we argue that the effect of remineralized terrigenous OC on radiocarbon-based ventilation age reconstructions may have gradually faded away towards the equator.

Reference:

Naafs, B. D. A., Hefter, J. & Stein, R. Millennial-scale ice rafting events and Hudson Strait Heinrich(-like) Events during the late Pliocene and Pleistocene: a review. *Qua. Sci. Rev.* **80**, 1-28 (2013).

Comments from Reviewer #2

This Manuscript investigates the role of terrestrial OC transported by IRDs on the estimation of deep ocean ventilation during Heinrich Stadial I in the North Atlantic Ocean. The authors use a wide array of OM-derived proxy records, like $\delta^{13}\text{C}$, $\delta^{15}\text{N}$, TOC%, C/N, and the ^{14}C age of (and corresponding ^{14}C age differences among) OC, planktic and benthic foraminifera. Analyses are performed in three cores located in the North and central Atlantic Ocean and retrieved within the IODP program.

The manuscript is well written, clear, and concise, and it reads very well. The theory to explain low ventilation ages at times of HS is novel and it delves into an aspect up to date overlooked by paleoceanographers. Despite the theory seems sound and plausible, the data, and their interpretation, are not entirely sufficient to provide robust evidence to support their hypothesis. In fact, I believe part of the presented data might be better explained by other mechanisms. Thus, there are some important aspects that should be addressed prior publication.

Response: We appreciate the positive assessment of our work and welcome your constructive suggestions! We provide a point by point response below and hope we succeeded in addressing the remaining concerns -

1. The authors ascribe the age offsets between OC and planktic foraminifera solely and uniquely to the addition of terrigenous OC. I read the whole Manuscript waiting for them to mention and discard other possible processes that might lead to similar age offsets, but they simply neglect them. Based on which data, estimates, and other bibliography do the authors disregard the potential addition of pre-aged and fossil OC resuspended from continental slopes (i.e., marine OC) and bioturbation?

Response: Thank you for pointing this important aspect out. We believe that other mechanisms such as the potential addition of pre-aged/fossil (marine) OC resuspended from continental slopes, bioturbation, and changes in sediment focusing/winnowing (Mollenhauer et al., 2011) may contribute to explain some aspects of the presented data, but the contribution of these effects are probably limited in this case study. We argue that OC much older (~ 4240 yr to 15440 yr) than co-deposited foraminifera particularly during HS1 and the YD, both of which were characterized by significant reduction in bottom current flow strength (Praetorius et al., 2008), must predominantly derive from continental sources. Although both corresponding author and the reviewer, Dr. Blanca Ausín, previously investigated the contributions of hydrodynamic processes on the fate of sedimentary OM in continental margins and OC aging (Bao et al., 2018, Ausín et al., 2021), such large offset may not be only interpreted, particular in a weakly-hydrodynamic deep ocean environment (Mollenhauer et al., 2011). Pre-aged OC remineralization would contribute the large ventilations which are based on radiocarbon results. In addition, the $\delta^{13}\text{C}$ and $\text{C}_{\text{org}}/\text{N}$ values suggest that the terrigenous OC were the

mainly carbon source in the North Atlantic during HS1. In the revised manuscript, we provide a more nuanced description and acknowledge the possible (yet arguably, secondary) contribution of pre-aged/fossil OC resuspended from continental slopes in potentially affecting radiocarbon-based ventilation age reconstructions. Hence, we add new sentence at Lines 107-110 as follows:

“Furthermore, our results indicate that the sedimentary OC is ~4,000 yr older than the coeval foraminifera at Site 1314, suggesting that pre-aged terrigenous OC may also have been transported to the northern North Atlantic (Fig. 2b)”.

Reference:

- Ausín, B., Bruni, E., Haghipour, N., Welte, C., Bernasconi, S.M., and Eglinton, T.I. Controls on the abundance, provenance and age of organic carbon buried in continental margin sediments. *Earth Planet. Sci. Lett.* **558**, 116759 (2021).
- Bao, R., van der Voort, T.S., Zhao, M., Guo, X., Montluçon, D.B., McIntyre, C., and Eglinton, T.I. Influence of Hydrodynamic Processes on the Fate of Sedimentary Organic Matter on Continental Margins. *Global Biogeochem. Cy.* **32**, 1420-1432 (2018).
- Mollenhauer, G., McManus, J. F., Wagner, T., McCave, I. N. & Eglinton, T. I. Radiocarbon and ²³⁰Th data reveal rapid redistribution and temporal changes in sediment focussing at a North Atlantic drift. *Earth Planet. Sci. Lett.* **301**, 373-381 (2011).
- Praetorius, S. K., McManus, J. F., Oppo, D. W. & Curry, W. B. Episodic reductions in bottom-water currents since the last ice age. *Nat. Geosci.* **1**, 449-452 (2008).

2. How do you explain the different magnitude of the OC-Planktic Foram age in cores that are at the same latitude and under the path of IRD belt? Specially, for BOF and 1308, which are located nearby? Measurement of duplicates, especially for core 1308, would be ideal.

Response: Thank you for pointing this out. A previous study indeed showed that the ¹⁴C ages of planktic foraminifera may have been affected by bioturbation and are hence much older than benthic foraminifera in the BOF core (Barker et al., 2004). Assuming that bioturbation also affects the planktic foraminifera ¹⁴C ages at nearby IODP site 1308, we would expect a similarly reduced OC-P age offset. However, in our study, we report a larger OC-P age difference (~ 4240 yr to 15440 yr) at IODP site 1308 when compared with the results reported in the BOF core. We thus conclude that the effect bioturbation may induce that the ¹⁴C ages of planktic foraminifera was reduced, if not absent, at IODP site 1308.

In addition, IODP core 1308 is located in the centre of the IRD belt, while IODP sites 1302 and 1314 are located at its edge (Hefter et al., 2017). We thus expect that more terrigenous OC was transported to the location of IODP site 1308, which may explain the difference in OC-P age

differences between the cores.

Reference:

Barker, S., Kiefer, T. & Elderfield, H. Temporal changes in North Atlantic circulation constrained by planktonic foraminiferal shell weights. *Paleoceanography* **19** (2004).

Hefter, J., Naafs, B. D. A. & Zhang, S. Tracing the source of ancient reworked organic matter delivered to the North Atlantic Ocean during Heinrich Events. *Geochim. Cosmochim. Acta* **205**, 211-225 (2017).

3. Site U1308 is the one presenting the largest age difference between OC and planktic foraminifera (ca. 14 kyr). Samples from Site U1308 are from hole B. As shown in Hodell et al., 2017, digital image analysis in U1308B evidenced large traces fossils and a high bioturbation index for HS1. Indeed, bioturbation has proven to have a large impact on the IRD record of U1308B by mixing the two distinct IRD peaks (H1.1 and H1.2.) observed in U1308A into one single peak during HS1. The authors overlook the potential impact of bioturbation on the data from this core. Bioturbation may partly explain (or even entirely?) the large observed age discrepancies.

Response: Thank you for pointing this out. Bioturbation may indeed play a non-negligible role in shaping the age difference between OC and planktic foraminifera in IODP core 1308 (and elsewhere). As the reviewer points out, this feature has previously been recognized by Hodell et al., 2017: “*the distinct double peak in Hole U1308A was primary in origin and not caused by upward dispersal of detrital carbonate from the lower peak by bioturbation. Moreover, the absence of a double peak in Hole U1308B is attributed to bioturbational disturbance that has mixed the two peaks.*”

Specifically, the largest age difference between OC and planktic foraminifera is reported from Section U1308B-1H-1 (84-86 cm) (i.e. 15.8 ka BP). The samples collected from U1308B correspond to the lower portion of the Heinrich Event 1 layer, essentially encompassing the same time interval as event H1.1 in U1308A (blue arrow, Figure below) for which the bioturbation index (BI) is decreasing. This indicates that the largest age difference between OC and planktic foraminifera that we report from U1308B coincides with the onset of Heinrich Event 1. As Hodell et al. (2017) emphasize, bioturbation would tend to mix younger sediment downwards, which may cause the larger age offset. We thus assume that even though bioturbation may affect the sediment deposits, our age offset estimates are conservative if anything, which does not challenge our initial conclusions.

Figure. Trace fossil distribution and bioturbation index (BI) in the studied Holes U1308A and U1308B (Hodell et al., 2017).

Reference:

Hodell, D.A., Nicholl, J.A., Bontognali, T.R.R., et al. Anatomy of Heinrich Layer 1 and its role in the last deglaciation. *Paleoceanography* **32**, 284-303 (2017).

4. In this regard, can you discard the impact of bioturbation on the other two cores? Additionally, sedimentation rates should be plot on age-depth model figures to identify periods of low sedimentation rate when the impact of bioturbation may increase.

Response: As mentioned above, while bioturbation may contribute to homogenize the sediments deposited during HS1 in the North Atlantic, its contribution may only be secondary in shaping the extraordinary ^{14}C age offset. We add a detailed illustration in support of the argument in the Supplement materials. In addition, we include temporal changes in sedimentation rates on the age-depth model figures at Extended Data Figs. 1, 3 and 4 as recommended. As the reviewer will reckon, the downcore sediment accumulation rates at cores U1302, 1308 and U1314 do not show substantial/abrupt changes across the last glacial termination, including HS1. And we add the discussion about the potential influence of hydrodynamic processes and bioturbation in the supplementary information.

“The potential influence of hydrodynamic processes and bioturbation

While OC aging driven by hydrodynamic processes has been reported on continental shelves^{8,9}, the pre-aging extent would be up to ~2,000 yr. However, it is difficult to interpret the extremely high ¹⁴C age difference between OC and coeval planktonic foraminifera (~ 4240 yr to 15440 yr) during HS1 considering only hydrodynamic processes. Prior study suggested that HS1 and YD may have been characterized by much weaker hydrodynamic conditions^{10,11}. In consequence, compared to co-deposited foraminifera (~ 4240 yr to 15440 yr) during HS1 particularly when considering the negative $\delta^{13}C_{org}$ values at Site UI308 (Fig. 2), the much older sedimentary OC would be more likely to due to enhanced input of pre-aged terrigenous OC to the North Atlantic. We thus consider that such large age offsets (~ 4240 yr to 15440 yr) may mainly arise from the contribution of pre-aged OC remineralization in the subsurface ocean, while OC aging related to hydrodynamical processes may also have played a secondary role in affecting radiocarbon ages.

In addition, bioturbation may affect the vertical OC age distribution in the sediment core¹². In our case, the largest age difference between OC and planktic foraminifera is reported from Section UI308B-1H-1 (84-86 cm) (i.e. 15.8 ka BP). Prior study has suggested that bioturbation decreased with depth during HS1 at this site¹². It thus implies that bioturbation would tend to mix younger sediment downwards, which may cause a larger age offset. We thus assume that even though bioturbation may have affected our sediment deposits, our age offset estimates remain conservative if anything, which does not challenge our conclusions.”

5. Line 124: “Microscopic evidence suggests that terrigenous OC is typically associated with ice-rafted debris (IRDs) (Extended Data Fig. 9)”. How can the authors tell from these pictures? OC (and its origin, i.e., marine or terrestrial) is not discernible from optical microscope pictures. In this line, IRDs have low Mineral Surface Area (MSA) because of their large size. This means that IRD grains host very little OC in comparison with smaller mineral grains having higher MSA. Such small grains can be laterally transported by deep currents, introducing way larger amounts of pre-aged OC, typically marine. The magnitude of the age discrepancy between OC and planktic forams is quite large at some sites, meaning that a large amount of pre-aged/fossil OC needs to be added to induce such bias. The question is, given that IRDs have very low MSA, can IRDs host and introduce the large amounts of OC needed to induce the observed age discrepancies? Related> Line 147: So anomalous that a huge amount of fossil OC would be needed to explain it. Again, can IRDs transport so much OC given their low MSA? Please check my comment above.

Response: Thank you for your comments. A previous study indeed suggested that OM is dominantly adsorbed on clay mineral surfaces as evidenced by the correlation between TOC and mineral Surface

Area (MSA, Mayer et al., 1994). However, we typically pick out larger rock fragments, characteristic of IRDs, from the pelagic sediment sample, not fitting the condition of the MSA hypothesis, given that measuring MSA through the BET methodology. The IRD sediments also contain fine particles. In addition, drifting icebergs may transport vast amounts of terrestrial OC to the IRD belt during HS1. According to Lambeck et al. (2014), global sea level may have risen by about 15m in a few thousand years during HS1. Considering a representative ocean volume and typical DOC concentrations reported in modern glaciers, we estimate that the iceberg armada may have transported more than 10 Pg DOC to the North Atlantic during HS1. Assuming that glaciers contain POC in concentrations about an order of magnitude higher than DOC (Hood et al., 2015), this implies that POC transported by drifting icebergs could amount to > 100 Pg. Thus, we believe the armada of drifting icebergs, characteristic of HS1 (and possibly the YD) may have deposited enormous amounts of pre-aged terrestrial OC during HS1, which would arguably have transiently affected the regional dissolved ^{14}C inventory. In addition, as shown in Extended Data Fig. 5, 6(g), 7(g) and 8(g), the $\delta^{13}\text{C}$ and the $\text{C}_{\text{org}}/\text{N}$ data shows that the primary source of sediments in the North Atlantic was terrigenous OC.

Reference:

- Lambeck, K., Rouby, H., Purcell, A., Sun, Y. & Sambridge, M. Sea level and global ice volumes from the Last Glacial Maximum to the Holocene. *Proc. Natl. Acad. Sci. U.S.A.* **111**, 15296-15303 (2014).
- Hood, E., Battin, T. J., Fellman, J., O'Neel, S. & Spencer, R. G. M. Storage and release of organic carbon from glaciers and ice sheets. *Nat. Geosci.* **8**, 91-96 (2015).
- Mayer, L. M. Surface area control of organic carbon accumulation in continental shelf sediments. *Geochim. Cosmochim. Acta* **58**, 1271-1284 (1994).

6. Line 131: the authors state that the ca. 4000 yrs age difference between OC and planktic forams at site U1314 suggests pre-aged terrigenous OC. As shown in Extended Data Fig.8, C isotope and C/N records do not fully support the primarily terrestrial origin of OC input during HS1 at that site. Moreover, these proxy records remain constant prior, during, and after that period. Why would any potential terrestrial OC input influence ventilation ages only during HS1 and not before or after? Accordingly, the observed age discrepancy in core U1314 could be ascribed to the lateral transport of marine OC. Why the authors do not even mention the potential impact of other factors like bioturbation or, more likely, the addition of pre-aged marine OC at this site? The corresponding author has implemented excellent prior work dealing on the contribution of hydrodynamic transport of marine OC to foraminifera-OC age discrepancies... Therefore, the data indicate that there is not enough evidence as to state that “Our data reveal that the old sedimentary OC deposited during HS1, both in the central (U1308 Site) and northern (U1314 Site) North Atlantic...”.

Response: Thank you for pointing this out. We apologize for the confusion. Owing to the lower sampling resolution at site U1314, Extended Data Fig.8 may not resolve the “peak” proxy values during HS1. However, the $\delta^{13}\text{C}_{\text{TOC}}$ values at U1314 fall within a narrow range (-24.22‰ to -24.01‰), diagnostic of terrestrial OC, consistent with Bard et al. (2000). We have revised our argumentation to avoid any unnecessary confusion (Lines 107-110):

“Furthermore, our results indicate that the sedimentary OC is ~4,000 yr older than the coeval foraminifera at Site 1314, suggesting that pre-aged terrigenous OC may also have been transported to the northern North Atlantic (Fig. 2b).”

About the terrestrial OC input influence ventilation ages with before and after HS1, previous studies stated that the Atlantic Meridional Overturning Circulation (AMOC) almost ceases during HS1 (Fig. 3i), which lead to a relatively-less exchange of younger ^{14}C -age carbon from the upper water mass with the bottom water mass than before and after the periods. It amplifies the effect of pre-aging organic carbon remineralization during HS1.

Furthermore, about the response to comment on the influence of bioturbation and hydrodynamic transport of OC on radiocarbon-based deep North Atlantic Ocean ventilation ages, we have replied previously at comments 1 and 3.

Reference:

Bard, E., Rostek, F., Turon, J.-L. & Gendreau, S. Hydrological Impact of Heinrich Events in the Subtropical Northeast Atlantic. *Science* **289**, 1321–1324 (2000).

7. Line 150: can such a small peak in sedimentary TOC concentration be considered as significant? Given the vigorous bioturbation affecting core U1308B during HS1, and that the TOC flux at that time is as low as before and after the event, can such low ventilation ages be associated with an input of terrigenous OC associated with IRDs?

Response: We welcome the opportunity to sharpen our argumentation. In that paragraph, our aim is to emphasize the spatial heterogeneity of the B-P ^{14}C ages. Here, we would like to emphasize the terrestrial OC contribution, IRD deposition, and ventilation change. We modified the sentences in updated manuscript as *“suggesting that the remineralization of old OC may be intensive, impacting on ventilation age reconstructions at site U1308.”*

About bioturbation affect, we have excluded its possibility of main contribution in the previous comments. Although we find that *“TOC flux at that time is as low as before and after the event”*, many studies found the enhanced supply of terrigenous OM during HS1 by means of specific biomarkers and sediment mass accumulation (Madureira et al., 1997; Manighetti and McCave, 1995; Rosell-Melé et al., 1997). The terrestrial OC proportion in the sedimentary OC at 1308 in Extended

Data Fig. 5 during HS1 is evidently peak. Additionally, this small peak was buried at almost twice the rate of the nearby thousands of years. In addition, the OC buried flux during the Holocene shown in Extended Data Fig. 7, shows a depth-related import trend of marine-source organic matter and rapid sedimentation during HS1, which may mute the “peak” of TOC flux, however, to sum up, the small peaks may imply a terrigenous OC input. And our updated manuscript mentions at Lines 130-134:

“While no significant increase in TOC burial was observed at Site U1308 during the deglaciation, enhanced supply of terrigenous OM has been reported previously based on biomarkers and reconstructions of past changes in sediment accumulation⁴⁷⁻⁴⁹, suggesting that the remineralization of old OC may be intensive, impacting on ventilation age reconstructions at site U1308.”

Reference:

Madureira, L. A. S. et al. Late Quaternary high-resolution biomarker and other sedimentary climate proxies in a Northeast Atlantic Core. *Paleoceanography* **12**, 255-269 (1997).

Manighetti, B. & McCave, I. N. Depositional fluxes, palaeoproductivity, and ice rafting in the NE Atlantic over the past 30 ka. *Paleoceanography* **10**, 579-592 (1995).

Rosell-Melé, A., Maslin, M. A., Maxwell, J. R. & Schaeffer, P. Biomarker evidence for “Heinrich” events. *Geochim. Cosmochim. Acta* **61**, 1671-1678 (1997).

8. Line 155: “We did not observe reduced ventilation during HS1 at Site U1302”. However, C isotope and C/N from U1302 do indicate high terrigenous OC input... Does this imply that the large input of terrestrial OC (as inferred from proxy records) at this site did not have any effect on ventilation ages but a smaller terrestrial input did it at other locations? The authors argue that this could be due to an increase in sedimentation rate, but they do not provide values.

Response: Thank you for pointing this out. This feature is consistent with high spatial heterogeneity in terrigenous OC remineralization. The somewhat lower sedimentary OC content at IODP site 1302 may be related to increased dilution (i.e. higher sediment accumulation), which led rapid accumulation of terrigenous OC. Furthermore, as shown in the figure below, the previous study has discussed the complex sources of IRD in the North Atlantic (Barker et al., 2015). The spatial variations of OC remineralization and consequent influences may be highlighted in the manuscript (Figure 3). In the updated manuscript, we add the sedimentation rates on age-depth model figures at Extended Data Fig. 1. And our updated manuscript mentions at Lines 137-139:

“We did not observe reduced ventilation as inferred by ¹⁴C age reconstructions during HS1 at Site U1302 (Fig. 1 and Extended Data Fig. 6), possibly owing to the high sedimentation rate (ranging between 16.5-17.3 cm/ka) and spatial differences of IRD sources⁵⁰.”

Redacted

Figure. Regional context of the study site. a, Modern sea surface temperature (SST) shown on colour scale in degrees Celsius. PF, polar front; AF, Arctic front. b, Major ocean currents. NAC, North Atlantic Current; IC, Irminger Current; EGC, East Greenland Current; NADW, North Atlantic Deep Water. c, d, Modern and LGM distribution of %NPS (shown on colour scale). Also shown are the locations of ODP site 983 (point 1) and ODP site 980 (point 2). (Barker et al., 2015)

Reference:

Barker, S. et al. Icebergs not the trigger for North Atlantic cold events. *Nature* **520**, 333-336 (2015).

9. Lines 183-185: The authors claim “In contrast, the high-latitude North Atlantic was characterised by generally more poorly ventilated conditions (with ~5,000 yrs of B-atm) at both abyssal (Fig. 3f) and intermediate (Fig. 3g) depths during HSI” But... AMOC stopped at that time, and ventilation of intermediate and deep water would decrease compared to the previous periods even without the input of terrigenous OC...

Response: Yes, A MOC decreased quite substantially during HSI owing to buoyancy forcing and as a result the ventilation rates of intermediate and deep water would decrease. Yet, previously reported ventilation age reconstructions can be up to hundreds even a thousand of years (Thornalley et al., 2011). This differs significantly from the poorly ventilated conditions (with ~5,000 yrs of B-atm) we observed in the IRD belt. We think that the contribution of pre-age OC remineralization can affect the estimated poorly ventilated conditions. The influence of terrestrial old OC would be a new mechanism triggering so-called the poor ventilations, which should not be negligible. Therefore,

ventilation reconstruction in the future may take into account both aged OC remineralization and real ventilation deterioration.

Reference:

Thornalley, D. J. R., Barker, S., Broecker, W. S., Elderfield, H. & McCave, I. N. The Deglacial Evolution of North Atlantic Deep Convection. *Science* **331**, 202–205 (2011).

10. Extended Data Fig. 5. A 90% contribution of terrigenous OC seems a lot for sites so distant from the continents. Did the authors account for the degradation of more labile (i.e., marine) OC with time? It might be that the initial % of terrestrial OC was much lower, but increased with time due to the preferential remineralization of marine OC. What is measured is the remaining OC. In fact, if the remaining buried OC is terrestrial, does not this mean that what was preferentially remineralized (and then influenced the radiocarbon signature of deep-water masses) was the more labile, marine OC?

Response: Thank you very much for the suggestion. Previous studies reported that the input of marine OC may accelerate the remineralization of terrigenous refractory OC (Guenet et al., 2010; Trevathan-Tackett et al., 2018; Ward et al., 2019). With the increase of labile organic matter supply, remineralization rates of the more recalcitrant organic matter often increase due to the priming effect. Additionally, refractory OC can be degraded by microbes and fungi in marine environments (Wang et al., 2020; Xu et al., 2018). This implies that marine and arguably more refractory terrigenous OC may be mineralized simultaneously, or possible that labile OC contributes to accelerate the remineralization of terrigenous pre-aged OC. These aspects are now explicitly mentioned, L. 183-187:

“Furthermore, marine-sourced, labile OC may be pre-aged in the water column before deposition⁵⁹. The aged marine OC degradation may further accelerate the remineralization of pre-aged terrigenous refractory OC^{60,61}. Considering the duration of the events, the maturation and degradation of marine OC and its potential impact on ventilation age reconstructions may be a target for research in the future.”

Reference:

Guenet, B., Danger, M., Abbadie, L. & Lacroix, G. Priming effect: bridging the gap between terrestrial and aquatic ecology. *Ecology* **91**, 2850-2861 (2010).

Trevathan-Tackett, S. M., Thomson, A. C. G., Ralph, P. J. & Macreadie, P. I. Fresh carbon inputs to seagrass sediments induce variable microbial priming responses. *Sci. Total Environ.* **621**, 663-669 (2018).

Wang, P. et al. Niche specificity and potential terrestrial organic carbon utilization of benthic Bathyarchaeota in a

eutrophic subtropic estuarine system. *Chem. Geol.* **556**, 119839 (2020).

Ward, N. D., Sawakuchi, H. O., Richey, J. E., Keil, R. G. & Bianchi, T. S. Enhanced Aquatic Respiration Associated With Mixing of Clearwater Tributary and Turbid Amazon River Waters. *Front. Earth Sci.* **7** (2019).

Xu, Y., Ge, H. & Fang, J. Biogeochemistry of hadal trenches: Recent developments and future perspectives. *Deep Sea Res. Part II Top. Stud. Oceanogr.* **155**, 19-26 (2018).

11. Why is the YD not discussed at all?

Response: Thank you for pointing this out. The forcing mechanisms(s) underpinning the YD are still debated. In addition, the YD event may bear considerably-wider spatial impacts on land, compared to HS1, there are thus more potential confounding variables and complex causes inherent to the YD. Thus, in this study, we focus on HS1.

12. Fig. 2 b: Data on cores U1308 and U1302 (the ones showing the largest OC-Planktic forams age discrepancies) are only available for HS1 and YD. Some data on other periods (e.g., Holocene, BA) would be desirable to see whether such large age discrepancies are characteristic of these periods or a relatively permanent feature of these sediments for the last deglaciation and Holocene.

Response: Thank you very much for the suggestion. Three sets of OC-P age determinations were measured for the Holocene and BA, yet the data do not reveal a significant contribution of terrigenous OC remineralization during the Holocene and BA. There is no evidence, to the best of our knowledge, in support of enhanced terrestrial OC input to the northern Atlantic during these intervals. Hence, we focus on the large, hitherto unrecognized large OC-P age differences during HS1 to highlight the contribution of pre-aged OC remineralization associated with the extraordinary ice rafting events. We believe this mechanism bear a long-standing impact on the regional radiocarbon inventory and by inference ventilation-age reconstructions. Further investigations are required to unravel the potential impact pre-aged OC remineralization bears on ventilation-age reconstructions during previous Heinrich events to assess whether this is a recurring phenomenon.

13. In summary, my major concerns are related to the fact that the authors do not provide robust evidence for some of the statements they make or some of the conclusions they reach. The presented data could be partly or entirely explained by the influence of other processes (e.g., bioturbation (consistent with the evidence existing from core U1308 (Hodell et al., 2017)) and lateral transport of marine OC (consistent with the data provided by the authors for core U1314)). However, the authors do not provide any estimates of the potential influence of such processes. More worryingly, they do not even mention these nor discuss them on the text. If the influence of this processes cannot be discarded, one cannot unequivocally and exclusively ascribe low ventilation ages to the input of

terrestrial OC.

Response: Thank you for your careful recommendations. We now acknowledge the potential impact of bioturbation and lateral transport of marine OC, in the revised version of our manuscript. Yet, we remain confident that the remineralization of pre-aged terrigenous OC may predominantly affect ventilation-age reconstructions, providing a testable hypothesis to account for the extremely old ventilation age (thousands of years) reported for HS1 and to a lesser degree, the YD. We believe that these aspects are novel and have been previously overlooked by paleoceanographers and climate modelers.

More in detail:

14. The authors claim reduced ventilation at Site U1314 and ascribe it to the input of terrigenous OC associated to IRDs, but terrigenous OC input seems constant in relation to previous and later periods according to various isotope proxy records.

Response: Thank you for pointing this out. We refer the reviewer to the response to questions 1 and 7 above.

15. The authors claim reduced ventilation at Site U1308, which, to the best of my knowledge, is heavily affected by bioturbation (Hodell et al., 2017).

Response: Amended. We refer the reviewer to the response to question 3 above.

16. The authors observe high ventilation at Site U1302, even negative values, which indicates something is affecting those ages (for which the authors give no satisfactory explanation). However, proxy records of terrigenous OC input at this Site do indicate an important input. They argue this could be due to high Sed. Rates, but values are not provided.

Response: Thank you for pointing this out. We refer the reviewer to the response to question 4 above.

Minor comments:

17. Line 50: Does this refer to the surface or the deep reservoir?

Response: We have replaced “high-latitude North Atlantic Ocean” with “high-latitude deep North Atlantic Ocean” at Line 47-48 to avoid confusion.

18. Lines 54-56: This notion could be better explained, please rephrase.

Response: We have rephrased the sentence (Line 53-56) as follows:

“When pronounced local ventilation anomalies are taken into consideration, such as ventilation ages exceeding 5,000 years in the intermediate-depth eastern equatorial Pacific, the dominant paradigm involving the advection of poorly ventilated water masses becomes, however, potentially inconsistent^{17,18}.”

19. Line 84: By “abnormal” do you mean “negative”? Please rephrase.

Response: We delete it.

20. Fig. 1 ii) I think you mean OC%, since $\delta^{13}\text{C}$ is not shown. Please, change the figure or the text “The terrigenous OC is indicated by the $\delta^{13}\text{C}_{\text{org}}$ data”.

Response: We have modified the text which now reads: “sedimentary terrigenous OC content (indicated by the $\delta^{13}\text{C}_{\text{org}}$ data) at sites U1302 (magenta), U1308 (green), and U1314 (blue)”.

21. Fig.3. “(data of Site SHAK06-5K (2.6 km, sky blue) and Site MD99-2334 (3.1 km, pink)”. Records are reversed, the blue one corresponds to MD99-2334 and the pink one corresponds to SHAK06-5K.

Response: Good catch. We have modified Figure 3 in the revised version of the manuscript.

22. Line 193: a word is missing. A poorly ventilated... (ocean, water mass?) in the North Atlantic.

Response: We have rephrased the sentence (Line 160-162) as follows:

“Would SSW have substantially intruded the deep North Atlantic during HSI, it would have resulted in ventilation ages (B-P ^{14}C ages) ranging between ~2,000 and 4,000 yr, which is arguably incompatible with a sluggishly ventilated high-latitude North Atlantic at that time.”

23. Lines 214-215: That is not true. The graph shows millennial-scale variations in Fig. 3 d-e. You may want to say that those variations, which undoubtedly exist, are of a smaller magnitude than those found in the north.

Response: Yes, indeed, the millennial-scale variations are of smaller magnitude in the north. We have rephrased at line 169-173 as follows:

“Furthermore, millennial-scale variations in ocean ventilation in the (sub)polar North Atlantic (Fig. 3f – g) and Arctic Mediterranean (Fig. 3h) coincided with those reported in the mid- (Fig. 3d – e) and low-latitudes North- (Fig. 3c) and South Atlantic (Fig. 3b), albeit to a much smaller magnitude. Such abrupt variations in ventilation ages challenge the mechanisms proposed to account for water mass advection from the south.”

24. Line 231-233: *Could the authors develop on this idea? I don't follow their reasoning here.*

Response: Sorry for the confusing discussion. We removed the sentence altogether.

25. Line 404: *which benthic species?*

Response: Foraminifera in the IRD belt are generally rare. We had to use mixed benthic species including *Cibicidoides wellerstorfi*, *Uvigerina* spp. and *Melonis* spp. for ^{14}C analysis.

26. 414-415: *Please update references by adding Verwega et al., 2021.*

Response: Thank you very much for the suggestion. We added the reference as suggested.

Reference:

Verwega, M.-T., Somes, C. J., Schartau, M., Tuerena, R. E., Lorrain, A., Oschlies, A., and Slawig, T. Description of a global marine particulate organic carbon-13 isotope data set. *Earth Syst. Sci. Data* **13**, 4861–4880 (2021).

- Blanca Ausín

Response: Thank you very much for your careful and constructive review of our manuscript.

Comments from Reviewer #3

Liu et al. presents compelling new evidence from organic carbon ^{14}C signatures from sediment cores located in the high latitude North Atlantic to explain the poorly ventilated deep North Atlantic indicated by ^{14}C from benthic foraminifera during the last deglaciation. Overall, the results are convincing and the manuscript is generally well-structured.

Response: We appreciate your efforts in reviewing our manuscript and the positive assessment of our work!

I: My major comment is about the term “ventilation”, which should be clarified in the manuscript. Ocean ventilation is controlled by the physical circulation/overturning rates and surface air-sea gas exchange. ^{14}C has been used to indicate ocean ventilation, but it is not a direct measurement of it. For example, the “reduced ventilation” in line 155 is actually “older ^{14}C age”. The definition of ventilation, ventilation age, and ^{14}C age should be described in the manuscript to avoid confusion.

Response: Thank you for pointing this out. We report ventilation ages as the ^{14}C age difference between benthic foraminiferal and the contemporaneous atmosphere (B-atm) to quantify past changes in North Atlantic deep-sea ventilation. In addition, we add now explicitly define what we mean by

ventilation, Lines 50-52:

“Reduced ventilation - defined here as the degree of air-sea equilibration - in the North Atlantic has commonly been attributed to dynamic changes in the Atlantic Meridional Overturning Circulation (AMOC) and the northward advection of poorly ventilated water masses from the Southern Ocean^{9,16}”

2: Furthermore, this study attributes the old ¹⁴C in the deep North Atlantic to the pre-aged terrigenous organic carbon. The authors claim that their results provide a unique perspective regarding ocean ventilation. However, this “perspective” is not well discussed in the manuscript. What is the ocean ventilation (physical circulation and air-sea gas exchange) during the deglaciation that can be inferred from the new OC 14C results? The current manuscript only explains the 14C age instead of ocean ventilation.

Response: Thank you very much for your comment. In fact, we aim to reduce errors in radiocarbon-based ventilation age reconstructions. And our work provides a new perspective for the reconstruction of ventilation ages associated with extreme climate intervals of the past, which have typically been overlooked by paleoceanographers and climate modelers.

Radiocarbon age offsets between contemporary benthic (B) and planktic (P) foraminifera (B-P ¹⁴C-age differences) have been used as a proxy for ocean mixing between (near)surface and bottom waters (Broecker et al., 2004). This ventilation proxy is often applied in regions where localized deep mixing exerts a significant influence on local deep-water radiocarbon activity. When independent calendar ages for marine records or knowledge about past surface reservoir ages are available, B-P age differences can be converted into Benthic-Atmosphere (here referred to as B-A) age differences. This allows for deep-ocean radiocarbon activities from contrasting hydrographic regimes to be correlated to a common atmospheric reference and provide a measure of the local deep ocean-atmosphere radiocarbon isotope disequilibrium (Skinner et al., 2010). In this study, we analyze the possible causes of poor ocean ventilation during HS1, and propose that pre-aged terrigenous OC supply and subsequent remineralization may significantly bias the ¹⁴C-age of benthic foraminifera. In the other aspect, our work would advance the understanding the ¹⁴C ages of foraminifera, e.g., ages model for the HS1 duration. Thus, this mechanism may contribute to significantly overestimate the age of the bottom water mass with consequences for our ability to model the forcing mechanisms underlying large changes in deep ocean ventilation.

References:

Broecker, W. et al. Ventilation of the Glacial Deep Pacific Ocean. *Science* **306**, 1169-1172 (2004).

Skinner, L. C., Fallon, S., Waelbroeck, C., Michel, E. & Barker, S. Ventilation of the Deep Southern Ocean and

Deglacial CO₂ Rise. *Science* **328**, 1147-1151 (2010).

3: Another comment is the usage of B-P ¹⁴C age to estimate ocean ventilation. Sites in this study are bathed by water sourced from the Southern Ocean. Using planktonic ¹⁴C ages from the North Atlantic will bias the estimation of the ventilation age. Will this affect the conclusion claimed in this study?

Response: Thank you for the suggestion. We exclude the effects of Southern Ocean water mass intrusion by comparing differences in ventilation ages at different latitudes. In Fig. 3, we show that high latitudes ventilation ages are older than those from low latitudes. Based on the data in the Extended data S2, the ¹⁴C age offsets between planktonic foraminifera and atmosphere CO₂ ranged from 373 yr to 979 yr (Excluded the condition of negative ventilation age). This suggests that the ¹⁴C age of planktonic foraminifera has aged to reduce the B-P ¹⁴C age about several hundred years by environmental factors, and we need to increase our ventilation age during the condition of B-P ¹⁴C age. In addition, we hereafter report ventilation ages as the ¹⁴C age difference between benthic foraminifera and the contemporaneous atmosphere (B-atm) in Fig. 3, and this reduces the contribution surface water reservoir age bears on ventilation age reconstructions. Previous evidence from phytoplankton suggested that the water source from the Southern Ocean cannot affect the SE Greenland (Hendry et al., 2016). We suggest that SSW have a small influence on the extremely old radiocarbon ventilation ages reported for the North Atlantic during HS1. Hence, there is a need to focus on the impact of pre-aged OC remineralization on radiocarbon-based deep North Atlantic Ocean ventilation ages. And we updated the title to “*Pre-aged terrigenous organic carbon biases ocean ventilation-age reconstructions in the North Atlantic*” to specify the geographical focus of our study.

References:

Hendry, K. R., Gong, X., Knorr, G., Pike, J. & Hall, I. R. Deglacial diatom production in the tropical North Atlantic driven by enhanced silicic acid supply. *Earth Planet. Sci. Lett.* **438**, 122-129 (2016).

4: Line 191-193: The authors rule out the SSW advection to the North Atlantic by comparing the ventilation ages between SSW and North Atlantic as the SSW is younger than the North Atlantic. It takes time for SSW to be advected to the North Atlantic, therefore the water age in the North Atlantic is still older than the Southern Ocean. So the older ventilation age in the North Atlantic than the Southern Ocean is still compatible with SSW being advected to the North Atlantic.

Response: Thank you for pointing this out. Our study indeed reveals the yet hitherto overlooked importance of organic carbon remineralization driven by the IRD supply during HS1. The influence of pre-aged terrigenous OC remineralisation may largely advance our understanding of North Atlantic Ocean ventilation age reconstructions. However, considering cold and warm period exchanges during

HS1 and the deterioration of ventilation with increasing latitude shown in Figure 3, the SSW advection to the North Atlantic has little impact on the period in which we reconstruct the ventilation age.

In a previous study, the deep-ocean radiocarbon pattern supports the notion of the bipolar seesaw mechanism: when the deep ocean was flushed by radiocarbon-rich NSW, Greenland warmed, and when NSW was replaced by SSW, Greenland cooled (Robinson et al., 2005). Although HS1 being conventionally considered as a cold interval, at IODP site U1308, the planktonic foraminifera assemblages are consistent with cold conditions occurring prior to and during H1.1. Toward the latter part of H1.1 moderate warming of surface water occurred and continued through the intervening period between H1.1 and H1.2 in core 1308 (Hodell et al., 2017). The sample characterized by extreme radiocarbon ventilation ages in this study is located in the midst of the transient warm interval, possibly contributing to the melting of the North American ice sheet. Assuming a complete invasion of the deep North Atlantic by SSW during HS1 (a feature that is currently hotly debated), it would be difficult to explain the alternating warm and cold intervals during HS1.

Therefore, the northward intrusion of SSW maybe not the main reason underpinning the reported large changes in ventilations ages. Rather, enhanced supply and remineralisation of pre-aged terrigenous OC transiently affected the regional radiocarbon inventory, contributing to the older reported ventilation age in the North Atlantic during HS1. Our updated sentence is at Lines 160-162 as follows:

“Would SSW have substantially intruded the deep North Atlantic during HS1, it would have resulted in ventilation ages (B-P ^{14}C ages) ranging between ~2,000 and 4,000 yr, which is arguably incompatible with a sluggishly ventilated high-latitude North Atlantic at that time.”

References:

- Hodell, D.A., Nicholl, J.A., Bontognali, T.R.R., Danino, S., Dorador, J., Dowdeswell, J.A., Einsle, J., Kuhlmann, H., Martrat, B., Mloneck-Vautravers, M.J., et al. Anatomy of Heinrich Layer 1 and its role in the last deglaciation. *Paleoceanography* **32**, 284-303 (2017).
- Robinson, L. F. et al. Radiocarbon Variability in the Western North Atlantic During the Last Deglaciation. *Science* **310**, 1469-1473 (2005).

Reviewer #2 (Remarks to the Author):

In this review, the authors have addressed all my concerns in their point-by-point reply.

Despite the revised manuscript shows no dramatic changes in the argumentation, I am satisfied the authors have softened some of the statements for which further evidence would be ideal.

This way, this work presents a novel theory and leaves room for discussion within our community, a notion that should be valued very positively.

In short, this is a very interesting and novel paper and it is worth of publication in this journal.

I recommend the authors to address the following minor comments just for clarity:

Lines 47-50: I assume the authors mean the surface reservoir, but please state it.

Line 100: "Positive C/N ratio". C/N ratio cannot be negative, I understand what the authors mean, but please rephrase it.

Regarding my previous comment on line 102 "Microscopic observations suggest that terrigenous OC is typically associated with ice-rafted debris (IRD) (Extended Data Fig. 9)". What I meant is that one cannot actually "see" OC on IRD surfaces under the microscope. Please rephrase when saying "microscopic observations suggest".

Lines 110-115: But why it is more reasonable? The authors do not explain. Is it because of the magnitude of the age discrepancies (= too large to be ascribed to bioturbation and lateral transport of OC)? Then please mention this, it may be clarifying for some readers.

Line 252-253: which type of species? Where they all epibenthic?

Response to Reviewers

We appreciate the constructive comments from the reviewers and editor concerning our manuscript entitled “*Pre-aged terrigenous organic carbon biases ocean ventilation-age reconstructions in the North Atlantic*” (Manuscript ID: NCOMMS-22-45880A). These comments are very helpful for improving our manuscript. In this point-by-point *Response to Reviewer 2*, we have addressed all the comments and suggestions. The responses to comments from reviewers and editor are listed below, with comments listed first in *blue italics*, followed by our response in black.

Comments from Reviewer #2

In this review, the authors have addressed all my concerns in their point-by-point reply. Despite the revised manuscript shows no dramatic changes in the argumentation, I am satisfied the authors have softened some of the statements for which further evidence would be ideal. This way, this work presents a novel theory and leaves room for discussion within our community, a notion that should be valued very positively. In short, this is a very interesting and novel paper and it is worth of publication in this journal.

Response: We appreciate the positive assessment of our work and welcome your constructive suggestions! We provide a point by point response below and we addressed the remaining concerns.

1. Lines 47-50: I assume the authors mean the surface reservoir, but please state it.

Response: Here we refer to the benthic-atmosphere radiocarbon age offset (B-Atm), and our updated manuscript mentions add: “*the benthic-atmosphere radiocarbon age offset*”.

2. Line 100: “Positive C/N ratio”. C/N ratio cannot be negative, I understand what the authors mean, but please rephrase it.

Response: We have replaced “Positive C/N ratio” with “maximum C/N ratio” at Line 97.

3. Regarding my previous comment on line 102 “Microscopic observations suggest that terrigenous OC is typically associated with ice-rafted debris (IRD) (Extended Data Fig. 9)”. What I meant is that one cannot actually “see” OC on IRD surfaces under the microscope. Please rephrase when saying “microscopic observations suggest”.

Response: We have rephrased the sentence as follows: “*Microscopic observations suggest that the sediment layer is enriched with ice-rafted debris (IRD) (Supplementary Fig. 9), indicating that OC is intimately related to ice-rafting events characteristic of HS^{15,16,33}.*”

4. Lines 110-115: *But why it is more reasonable? The authors do not explain. Is it because of the magnitude of the age discrepancies (= too large to be ascribed to bioturbation and lateral transport of OC)? Then please mention this, it may be clarifying for some readers.*

Response: Yes, thank you for your comments and suggestion. Indeed, it is because the age differences we observed are too large to be described by bioturbation and lateral transport of OC. And we have rephrased the sentence as follows: “*it is difficult to interpret such large ¹⁴C age difference between OC and coeval planktonic foraminifera (~ 4240 yr to 15440 yr) during HS1 in northern North Atlantic.*”

5. Line 252-253: *which type of species? Where they all epibenthic?*

Response: We have rephrased the sentence as follows: “*The benthic foraminifera for conducting ¹⁴C analysis included Cibicidoides wellerstorfi, Uvigerina spp. and Melonis spp.*”